# Multi-Level Feature Fusion in CNN-Based Human Action Recognition: A Case Study on EfficientNet-B7

**DOI:** 10.3390/jimaging10120320

**Published:** 2024-12-12

**Authors:** Pitiwat Lueangwitchajaroen, Sitapa Watcharapinchai, Worawit Tepsan, Sorn Sooksatra

**Affiliations:** 1National Electronic and Computer Technology Center, National Science and Technology Development Agency, Khlong Luang, Pathum Thani 12120, Thailand; pitiwatacademic@gmail.com (P.L.); sitapa.wat@nectec.or.th (S.W.); 2International College of Digital Innovation, Chiang Mai University, Mueang Chiang Mai, Chiang Mai 50200, Thailand; worawit.tepsan@cmu.ac.th

**Keywords:** human action recognition, fusion method, multi-level fusion

## Abstract

Accurate human action recognition is becoming increasingly important across various fields, including healthcare and self-driving cars. A simple approach to enhance model performance is incorporating additional data modalities, such as depth frames, point clouds, and skeleton information, while previous studies have predominantly used late fusion techniques to combine these modalities, our research introduces a multi-level fusion approach that combines information at early, intermediate, and late stages together. Furthermore, recognizing the challenges of collecting multiple data types in real-world applications, our approach seeks to exploit multimodal techniques while relying solely on RGB frames as the single data source. In our work, we used RGB frames from the NTU RGB+D dataset as the sole data source. From these frames, we extracted 2D skeleton coordinates and optical flow frames using pre-trained models. We evaluated our multi-level fusion approach with EfficientNet-B7 as a case study, and our methods demonstrated significant improvement, achieving 91.5% in NTU RGB+D 60 dataset accuracy compared to single-modality and single-view models. Despite their simplicity, our methods are also comparable to other state-of-the-art approaches.

## 1. Introduction

Human action recognition (HAR) is a key area of computer vision focused on detecting, analyzing, and interpreting human movements and actions. Precise HAR models are vital for a range of applications, including human behavior analysis, autonomous driving, sports analytics, and healthcare [1,2,3,4,5,6,7]. One effective strategy to improve HAR model performance is by incorporating additional input data or combining multiple modalities, such as skeleton data, depth maps, or point clouds, rather than relying solely on a single source of information, such as RGB frames. However, optimally combining multiple modalities, such as RGB images and skeleton data or skeleton data and point clouds, remains a complex challenge, especially due to the heterogeneous nature of these data sources.

Using multimodal approaches can help overcome the common challenges commonly found in using a single data source. For instance, when using RGB information, issues such as varying lighting conditions and background clutter can introduce noise and thus lower the model performance. Likewise, skeleton-based methods may suffer from issues such as fast object movement and inaccurate joint detection [8], while unimodal models offer advantages like lower complexity, simpler architecture, and the ability to operate with a single data source, they also have inherent limitations [9]. For instance, RGB frames lack 3D structural information, and skeleton data may miss contextual features. Consequently, unimodal approaches often struggle to achieve highly precise models.

An early approach to combining information from different data streams was the two-stream architecture proposed by Simonyan et al. [10]. This method integrates spatial and temporal information through late fusion, a technique that typically combines features or decision scores at the final stage of processing. This approach utilizes spatial information from RGB frames and temporal information derived from optical flow (OF) images. In the late fusion part, the softmax output from each stream is summed together. This strategy is simple and notably outperformed other models from that time that relied solely on a single stream of information. In the bigger perspective, two different types of modalities apart from RGB and OF frames can be fed to the model. For example, one can integrate different data types like RGB frames and skeleton data, or skeleton data and point clouds. If using a two-stream architecture, once the softmax is extracted from each stream, these softmaxes will be combined with the predicted class based on their maximum confidence in the summed components.

From the perspective of feature fusion, two-stream models [10] and recent triple-stream models, such as MMnet [9], have primarily relied on late fusion, which is a basic form of ensemble learning [9], while late fusion offers some performance gains, depending solely on it can limit the model’s ability to fully exploit the richness of multimodal data. A more advanced approach involves fusing features at multiple stages in the network, rather than just at the final layer. By combining information earlier and more dynamically, models can better capture intricate relationships between modalities. In fact, Feichtenhofer et al. [11] demonstrated that incorporating even simple intermediate fusion, which simply implies fusing features in a middle layer of the networks, in addition to late fusion, significantly improves accuracy in the original two-stream model. This illustrates the value of integrating features across various levels of the network, rather than restricting fusion to the final layer.

A natural question that arises is how to establish communication between networks that process each modality. This would help the model to effectively exploit information from each stream. One simple method is to fuse the modalities before they enter the feature extraction process, a technique known as early fusion [12]. In early fusion, the raw data from different modalities are combined into a unified representation prior to feature extraction. Another method is by creating connections between the networks that process features from each stream, resulting in a more unified network compared to architectures that rely solely on late fusion. The technique of combining features at multiple levels is commonly referred to as multi-level fusion [13,14]. This approach has been applied in various applications, such as image dehazing [13] and HAR in healthcare [14], both of which have shown impressive performance. In unimodal models, this technique can refer to the reuse of low-, mid-, and high-level features extracted in the same model, a technique commonly used in convolutional neural network (CNN)-based architectures such as ResNet [15] and DenseNet [16]. However, our research focuses on combining multimodal features using multi-level fusion, facilitating inter-model connections at multiple levels.

However, a key challenge in multi-level fusion with a large number of modalities is that the number of possible inter-model connections grows exponentially, particularly in deep networks where layers from different models can be fused. An additional challenge arises when different types of data require different models for processing. For instance, RGB frames might be processed using a CNN-based model, while skeleton data may require a graph-based model. This further increases the complexity, as the number of potential ways to combine information from both networks grows exponentially, making it difficult to identify optimal connection points.

In real-world scenarios, using multiple modalities is often impractical due to constraints such as limited time, budget, or data availability. As a result, effectively leveraging features from a single data source becomes a more viable solution. In our research, we applied the principles of multimodality and multi-level fusion while relying exclusively on RGB frames as our single data source. Although the NTU RGB+D dataset includes 3D skeleton data, we intentionally excluded these to ensure our model can be applied in practical settings without the need for additional sensors to collect additional data.

Specifically, we processed RGB frames to extract both OF and skeleton information through a multimodality approach. This enhances the model by integrating region of interest ROI-based OF and ROI-based RGB data during training, enabling us to deploy the model without the need for extra data collection or specialized devices, such as 3D skeleton sensors. Specifically, while only RGB frames are directly available, we leveraged a pre-trained model to generate OF frames and extract 2D coordinates for cropping ROIs of human body parts in the video frames. This provides the model with multiple perspectives: RGB frames offer spatial information, while OF frames capture temporal dynamics through movement patterns.

We demonstrate that the multi-level fusion technique is a promising method for improving performance in HAR, even when starting with a single data source. This is evidenced by the significant performance improvement compared to standard techniques applied to a single data source.We present a case study on the use of multi-level fusion (MLF) within EfficientNet-B7 (EffB7) for HAR, integrating early, intermediate, and late fusion within the model.For practical applications, we developed a method to leverage multimodality using preprocessing from a single data source, making the approach more feasible in real-world scenarios.

## 2. Related Work

We begin by discussing data fusion methods as applied to video classification. Fusion methods are commonly used to combine features from different inputs with the goal of obtaining consistent, informative, and accurate data. This process typically leads to enhanced accuracy, improved robustness, and greater completeness, as the fusion of features creates a unified representation that allows the model to better capture the relevant information from the original inputs [17]. Fusion techniques have also been widely applied across various fields, including remote sensing, medical imaging, and robotics, to address challenges such as data incompleteness, conflicts, or inconsistencies in raw data collections [18,19]. Based on the location or level at which information is fused [12], fusion methods in video classification can be broadly categorized as follows:Early fusion: At this level, features are typically fused at the beginning of the network, or before they enter the network. This approach involves processing information in advance, such as combining sensor channels into a unified representation.Intermediate fusion: Intermediate fusion occurs in the middle layers of the network, where features extracted from each modality or source are combined. This method integrates both high and low-level features from different modalities, providing a more comprehensive representation of the combined features.Late fusion: At this level, the output confidence scores from different classifiers are fused, and the combined outputs are used by the classification system to make the final decision. These scores can also be utilized as input for training machine learning models, such as Support Vector Machines (SVMs) [20], which often result in improved accuracy [10,11].

Depending on the field and factors such as the type and quality of input data, fusion methods often improve the accuracy and robustness of systems. However, this is not always the case, and further research is needed to fully understand their effectiveness.

### 2.1. Early Approach to Applying Fusion in HAR-Based Models

An early approach to fusing different modalities was introduced in the two-stream model [10], which unified temporal and spatial information from video frames and OF images, presenting spatio-temporal features to the model. In this approach, spatial features were extracted from individual RGB frames, while temporal information was derived from a sequence of OF displacement fields calculated between consecutive frames using a convolutional network. The resulting softmax scores from both streams were then combined using late fusion methods, such as score summation or SVMs, to make the final classification decision.

A more recent enhancement of this two-stream architecture [21] improved performance by incorporating long short-term memory (LSTM) networks [22] to process each frame. The outputs were then merged in the late fusion stage, resulting in higher classification accuracy.

### 2.2. The Introduction of Intermediate Fusion in CNN-Based HAR Models

Later, Feichtenhofer et al. [11] introduced new methods for fusing data at intermediate layers of the previously proposed two-stream model, going beyond the conventional late fusion approach. These methods emphasize combining spatial and temporal features during the training process, leading to more effective learning of spatio-temporal representations. One proposed intermediate fusion technique uses an end-to-end training approach that processes both streams into one unified stream simultaneously, using two connections at the intermediate fusion between the two networks. This fusion takes place at intermediate stages using techniques such as sum fusion, max fusion, concatenation, and convolutional fusion, which will be discussed in detail in Section 3.3.2.

Another method proposed in [11] treats the spatial and temporal streams as two independent components. In this approach, the spatial and temporal features extracted from the two networks are fused at an intermediate layer of the temporal stream, forming a spatio-temporal stream. Meanwhile, the spatial features continue through the spatial stream toward the final softmax layer. As a result, instead of having separate spatial and temporal streams, we now have a spatio-temporal stream alongside a spatial stream. Softmax scores are then extracted from both streams, and late fusion is applied using methods such as summing or machine learning models such as SVM to make the final classification decision.

This study demonstrated that both fusion methods, especially if both are applied, typically lead to increased accuracy, as they allow for better integration of spatial and temporal information throughout the extraction and classification processes.

Beyond the multimodal fusion point of view, these intermediate fusion methods can be viewed as a way to harness existing information within the network. During feature extraction, intermediate features, or snippets, can be transferred between streams. For instance, snippet features from the spatial stream can be fused at various layers of the temporal stream, and similarly, features from the temporal stream can be integrated at different layers of the spatial stream.

### 2.3. Recent Fusion Methods Applied to HAR

In a recent fusion method applied to HAR, Song et al. [23] proposed an approach to fuse multiple modalities effectively while focusing on human body parts. This method uses aggregated region of interest (ROI) images of body parts, along with global body crops derived from skeleton data. Body parts are cropped from both RGB and OF frames, and the aggregated patches, together with skeleton data, are processed through a ConvNet, followed by a stack of LSTM layers, forming triple streams to capture temporal information across frames. In the final stage, a sum is applied for late fusion to make the classification decision. Similarly, the recent triple-stream MMNet architecture proposed by Yu et al. [9] employs a method where cropped body parts from RGB frames are arranged into a single image and passed through a pre-trained model, without using LSTMs. In this case, the joint and bone data derived from the skeleton are used as two additional streams, and softmax scores are fused in the final layer by summing.

In our approach, we propose using early, intermediate, and late fusion within a unified architecture using only RGB video data, aiming for effective accuracy without relying on additional information such as depth images or 3D skeleton joints. Inspired by the work of Yu et al. [9], we employ a similar technique to extract ROI RGB frames and ROI OF images. We then apply early fusion by combining both ROI RGB and ROI OF images, enriching the OF images with additional information from the RGB frames.

## 3. Methodology

We begin with an overview of the proposed architecture in Section 3.1. The model components are then described in detail, with the spatial and temporal streams outlined in Section 3.2. Finally, the multi-level fusion strategy is introduced in Section 3.3.4.

### 3.1. Overview of the Proposed Architecture

In this section, we present an architecture that effectively combines spatial and temporal information using both multi-level and multimodal approaches. As illustrated in Figure 1, the proposed architecture consists of two independent streams, namely, a temporal stream and a spatial stream, referred to as Model 1 and Model 2, respectively. Model 1 and Model 2 can be pre-trained models, and they do not need to be identical; each model can be selected based on its effectiveness in extracting features from its respective input modality.

In our architecture, features from each data stream are extracted using Model 1 and Model 2 for the temporal and spatial streams, respectively. To extract intermediate features, each model is divided into two parts: the first part captures intermediate features, while the combined features (or the original features from the stream) are passed to the second part, which constitutes the remainder of the model. As shown in Figure 1, intermediate fusion in our design occurs within Model 1, the temporal stream. Although intermediate fusion can take place at multiple points within a model, it is applied at a single location in the temporal stream in our case.

The temporal stream utilizes early fusion to incorporate additional information from ROI-based OF, enhancing the representation of temporal data. In contrast, the spatial stream processes ROI-based RGB frames without applying early fusion. During the feature extraction process, intermediate fusion is applied in the temporal stream, which is then used to derive the softmax score for this stream. Similarly, a softmax score is obtained from the spatial stream. This architecture effectively fuses information through early, intermediate, and late fusion strategies.

In addition to the multimodal approach, temporal and spatial information along with multi-level fusion techniques are also incorporated into the model. The multi-level fusion process involves early, intermediate, and late fusion. Early fusion is only applied in the temporal stream, as mentioned.

In the intermediate fusion section, as illustrated in Figure 1, the processed ROI-based early-fused features are first extracted by the first part of Model 1, which consists of a partial block selected before the full extraction process is completed. Next, features extracted from the first part of Model 2, operating in the spatial stream, are combined using the intermediate fusion method. These fused features are then passed through the second part of Model 1 to obtain the softmax score for the temporal stream. Similarly, the softmax score for the spatial stream is also extracted from ROI-based RGB. Finally, late fusion is applied to combine the softmax scores from both streams, producing the final result. We note here that Model 1 and Model 2 can be different, and the first part from Model 1 and Model 2 can be varied.

### 3.2. Temporal and Spatial Streams

The temporal stream utilizes ROI-based OF generated from pairs of RGB video frames to extract temporal information. An example of a processed ROI OF image is shown in Figure 2. The prepared ROI-based OF are then resized into 224 × 224 pixels and fed to Model 1. Examples of the model are VGG-16, ResNet-18, or EfficientNet-B7.

We utilize ROI-based OF to extract temporal information, as it captures the pattern of apparent motion between frames, resulting from the relative movement between the observer and the scene. This method effectively facilitates the acquisition of temporal data. OF has been widely employed in applications such as video compression, motion estimation, and action recognition, providing a compact representation of motion within video sequences. In our approach, OF is generated by inputting pairs of RGB frames into RAFT (Recurrent All-Pairs Field Transforms) [24], a recent end-to-end deep learning model known for its high accuracy and low computational time in OF estimation.

Similar to the temporal stream, the spatial stream uses ROI-based RGB, as depicted in Figure 2, to capture spatial information. The ROI RGB inputs are also resized to 224 × 224 pixels and fed to Model 2. This processing begins with the initial layers of Model 2, without any intermediate fusion from the corresponding layers of Model 1. The output then passes through the remaining layers of Model 2. The softmax score for this spatial stream is then extracted at the decision layer of this stream.

### 3.3. The Fusion Methods

We first describe the fusion methods applied in our proposed model, comprising early, intermediate, and late fusion. These fusion methods are also applied in both streams together to leverage the multi-layer fusion technique.

#### 3.3.1. Early Fusion

Based on Figure 2, the top row shows that the processed OF frames often lack sufficient information in scenarios involving two or more people. This occurs because individuals with minimal movement in a video frame may not appear clearly in the OF frames, resulting in potential loss of information when multiple people are present. To address this issue, the early fusion in the temporal stream is employed to supplement the OF frames with additional data. Early fusion helps retain more comprehensive information by combining ROI RGB and OF. We applied the early fusion using the equation below:(1)$$I_{T}=\alpha I_{RGB}+\left(1-\alpha \right)I_{OF},$$
where α∈(0,1). Setting α close to 1 would mean that the temporal information originating in the OF images can be diluted by the RGB information, as shown in the $\left(1-\alpha \right)I_{OF}$ term. The notations $I_{T},I_{RGB},I_{OF}$ denote the early-fused ROI-based OF, ROI-based RGB, and ROI-based OF, respectively.

#### 3.3.2. Intermediate Fusion

We investigated the effects of intermediate fusion using common fusion methods proposed in [11]. Intermediate fusion was applied by combining spatial and temporal information at an intermediate layer within the temporal stream, as illustrated in Figure 1. After fusion, the combined features were further processed through the remaining layers until the classification stage. This approach can be regarded as a combined stream or a spatio-temporal stream. Specifically, a fusion function *f* is defined as:(2)$$f:x_{t}^{a},x_{t}^{b}⟶y_{t},$$
where $x_{t}^{a}$, $x_{t}^{b}$ are the feature tensors with size (H,W,D) and $\left(H,W,D^{\prime }\right)$, respectively, where *H* is the height, *W* is the width, and *D* or ($D^{\prime }$) is the depth of the feature maps. The output tensor $y_{t}$ is the output tensor after the fusion function is performed with size $\left(H,W,D^{\prime \prime }\right)$. In other words, a feature tensor $x_{t}$ with size (H,W,D) can be denoted by $x_{t}={\left(x_{i,j,k}\right)}_{t}$, where 0≤i≤H, 0≤j≤W, and 0≤k≤D. The fusion function that is performed at an intermediate layer can be defined as follows:Sum fusion: $x_{i,j,k}^{\mathrm{sum}}=x_{i,j,k}^{a}+x_{i,j,k}^{b}$ whenever $D^{\prime }=D^{\prime \prime }$.ax fusion: $x_{i,j,k}^{\max}=\max\left\{x_{i,j,k}^{a},x_{i,j,k}^{b}\right\}$ whenever $D^{\prime }=D^{\prime \prime }$.Concatenation fusion: $x_{i,j,2k}^{\mathrm{cat}}=x_{i,j,k}^{a}$ and $x_{i,j,2k-1}^{\mathrm{cat}}=x_{i,j,k}^{b}$.Convolution fusion: $x_{\mathrm{conv}}=x^{\mathrm{cat}}⊛f+b$ whenever $D^{\prime }=D^{\prime \prime }$, where *f* is a convolutional 2D filter with 1×1 kernel and a bias *b*.

These intermediate fusion strategies enhance the model by integrating spatial and temporal features, ultimately improving performance in the classification stage.

#### 3.3.3. Late Fusion

The late fusion method is utilized to enhance decision-making by averaging the softmax scores from the spatio-temporal and spatial streams.

#### 3.3.4. Multi-Level Fusion

Fusion can be performed at various levels within a network to maximize the feature quality during the extraction and learning processes [12]. Specifically, early fusion is implemented at the input level, intermediate fusion at the low-level features, and late fusion at the high-level features. In detail,

Early Fusion: This is implemented at the input level, where raw data (such as RGB frames and OF) from different modalities are combined before feature extraction begins. Early fusion ensures that shared patterns or correlations across modalities are considered right from the input stage.Intermediate Fusion: This occurs at the low-level feature stage, where features extracted by initial layers of the network are combined. At this level, intermediate fusion allows the integration of both spatial and temporal information or other feature representations before they pass through deeper layers, enabling a richer feature representation.Late Fusion: Performed at the decision-making level, where outputs such as softmax scores from different network branches are aggregated. This approach focuses on merging high-level semantic information to improve classification or recognition accuracy, typically using methods such as weighted averaging, concatenation, or ensemble learning techniques.

The proposed multi-level fusion (MLF) technique allows the network to leverage feature fusion at various stages of the learning process. The MLF mechanism is structured in three steps. First, the spatial stream, using ROI-based RGB, and the temporal stream, using early-fused OF-based ROI, are trained separately. Second, the spatio-temporal stream is formed by performing intermediate fusion at an intermediate layer in the temporal stream, utilizing information extracted from partial blocks of both the spatial and temporal streams. The extraction process then continues until the end of the temporal stream. Finally, late fusion is performed by averaging the softmax scores from the spatial and spatio-temporal networks to make the final decision.

## 4. Experimental Results

The proposed method was evaluated by performing experiments on the NTU RGB+D 60 and NTU RGB+D 120 datasets [25,26]. This section describes the datasets, data preprocessing, implementation details, ablation study, and comparative results.

### 4.1. Dataset

The NTU RGB+D 60 and 120 datasets: NTU RGB+D 120 dataset [25,26] is a large-scale action recognition dataset with RGB videos and 3D skeleton data. This dataset contains 114,480 video samples belonging to 120 different action classes. The actions in this dataset are in three major categories: single-person daily actions, two-person interactions, and medical conditions.

In Training and Testing protocols, we provide detailed descriptions of the training and testing protocols for both NTU RGB+D 60 and NTU RGB+D 120 datasets:

#### 4.1.1. NTU RGB+D 60

Cross-Subject (CS) Evaluation: The dataset comprises 40 unique subjects aged 10 to 35, representing diversity in age, gender, and height. Each subject is assigned a consistent ID throughout the dataset, enabling standardized and reliable evaluation.

Cross-View (CV) Evaluation: Actions were recorded using three cameras positioned at horizontal angles of −45°, 0°, and +45°. Each subject performed every action twice—once facing the left camera and once facing the right camera—offering diverse perspectives for robust and comprehensive evaluation.

#### 4.1.2. NTU RGB+D 120

Cross-Subject (CS1) Evaluation: The dataset consists of 106 subjects, split into two groups: 53 for training and 53 for testing. Pre-assigned subject IDs ensure consistency and comparability across various evaluation methods.

Cross-Setup (CS2) Evaluation: The dataset is divided based on setup IDs, with even-numbered setups allocated for training and odd-numbered setups for testing. This approach provides 16 setups for training and 16 setups for testing, ensuring balanced evaluation.

For further details on each evaluation protocol, please refer to the official documentation of the NTU RGB+D 60 dataset [25] and the NTU RGB+D 120 dataset [26].

### 4.2. Data Preprocessing

We first selected 20 frames from the video sequence using uniform sampling. YOLOV5-Pose [27] was then used to estimate the coordinates of key body points: the head, left hand, right hand, left foot, and right foot. These coordinates were used within each frame to crop the ROI, and the ROIs from all 20 frames were vertically concatenated. Subsequently, five vertical ROI images were uniformly selected and concatenated horizontally, after which they were resized to 224 × 224 pixels.

Next, two consecutive frames from the selected 20 video frames were used to compute the OF using RAFT (Recurrent All-Pairs Field Transforms for OF) [24]. The OF images were transformed using an inverted color scheme to eliminate white background pixels and then converted to grayscale, making only the moving body parts remain visible as white pixels. Five OF frames were uniformly selected to construct the ROI-based OF images. Using the coordinates from the RGB frames, we generated the ROI OF images following the same method as for the RGB images. In videos featuring two actors, each person’s region was cropped separately in each frame, divided in half, and horizontally concatenated with the region from the other person. The resulting ROI images are illustrated in Figure 2.

### 4.3. Implementation Details

The adaptive moment estimation (ADAM) optimizer was used for the training process, with an initial learning rate of 0.0003. The training was conducted for 30 epochs, with the learning rate reduced by a factor of 0.6 every 6 epochs. The batch size was set to 64.

### 4.4. Ablation Study

We assessed the model of individual components within our proposed framework through ablation studies. Initially, we analyzed the performance of the single streams independently; these were spatial streams using ROI-based RGB and temporal streams using early-fused ROI-based OF.

Subsequently, we conduct experiments to evaluate intermediate fusions. All experiments are evaluated on the NTU RGB+D dataset with a cross-subject (CS) protocol.

#### 4.4.1. Spatial Stream and Early-Fused Temporal Stream Evaluation

In this experiment, a single-stream network was evaluated using several pre-trained models, as shown in Table 1. The temporal and spatial streams were assessed independently. The temporal stream utilized an early-fused ROI-based OF image, while the spatial stream employed an ROI-based RGB image.

To narrow down the options for multi-level fusion, we first conducted experiments to identify the promising candidates for a multi-level fusion case study. The results indicate that EfficientNet-B7 [28] is the top performer for both the spatial and temporal streams, with ResNet-18 [15] ranking second in both streams.

**Table 1 jimaging-10-00320-t001:** Comparison of accuracy across different models for a single spatial stream (RGB) and a single temporal stream (early-fused OF) using the cross-subject protocol (CS).

Model	RGB	Early-Fused OF
VGG-16 [29]	73.38	70.57
DenseNet-121 [16]	72.68	72.23
ResNet-18 [15]	83.39	80.06
EfficientNet-B7 [28]	87.97	84.52
Swin-Transformer-B [30]	76.29	75.52
ViT-B Patch 16 [31]	71.07	65.40

#### 4.4.2. Intermediate Fusion

Focusing on the intermediate layers of EffB7, we evaluated the performance when both streams were combined at an intermediate layer, forming a spatio-temporal network. This was accomplished by applying simple intermediate fusion methods to merge features extracted from a partial block of the spatial stream (the first part of Model 2, as depicted in Figure 1) with features extracted from the first part in Model 1 (the first part of Model 1 in Figure 1).

In Figure 1, the partial block represents the first section of each main model after dividing Model 1 and Model 2. In this experiment, we selected EffB7 for both the spatial and temporal streams based on its top performance, as indicated in Table 1. Intermediate fusion was applied within the temporal stream by integrating features extracted from the partial blocks of both the spatial and temporal streams. Following this, the combined features were carried out until the end of the temporal stream.

We evaluated several common intermediate fusion methods, including sum, max, concatenation, and convolution, as well as different fusion points within EfficientNet-B7. The results are presented in Table 2. In the table, (S) and (T) refer to the spatial and temporal streams, respectively. Additionally, (i) and (j) denote the *i*-th and *j*-th blocks in EfficientNet-B7, as implemented using the PyTorch framework [32]. The *i*-th block corresponds to the *i*-th sequential layer, which contains several MBConv blocks within EfficientNet-B7. The structure used for EfficientNet-B7 follows the PyTorch implementation [32].

As the results in Table 2 show, when intermediate fusion methods are applied at different block levels, the EffB7(S)(6) and EffB7(T)(6) combination consistently achieves the highest accuracy across all fusion methods, except for concatenation. Among the fusion techniques, convolution consistently outperforms the others, achieving the highest accuracy of 88.46% for the block 6 combination.

The max fusion method also performs competitively, reaching 86.93% accuracy for the same block combination. In contrast, the sum and concatenation methods tend to deliver lower accuracy, with the sum method producing the lowest scores in most block-level combinations. The weakest performance is observed for the sum fusion applied at block 4, with an accuracy of 82.03%.

Accuracy tends to improve as features from deeper blocks (e.g., block 6) are fused, particularly for the convolution and max methods, suggesting that intermediate layers closer to the end of the network contain more informative features for fusion. Early block combinations, such as block 4, generally perform worse across all fusion methods, indicating that features from earlier layers may not be as discriminative or suitable for fusion.

Overall, block 6 emerges as the optimal fusion point, especially for the convolution method, which achieves the highest accuracy. Conversely, block 4 yields the weakest performance, particularly for the sum method.

### 4.5. Results and Discussion

In this section, we compare our proposed method with other state-of-the-art models. Our proposed method uses late fusion to combine spatio-temporal and spatial streams together, with EfficientNet-B7 as the backbone model. We evaluate performance on the NTU RGB+D 60 dataset, following cross-view (CV) and cross-subject (CS) protocols [25], as shown in Table 3. In the table, “3D-Pose” refers to the use of 3D skeleton data, while “RGB” indicates the use of RGB frames. We specifically compare models trained using a single-view approach. Models trained in multi-view approaches, which require multiple cameras for data collection, are not included, as they do not align with our objective of practical usage.

Table 3 presents a comparison of accuracy between our proposed MLF architecture and several state-of-the-art models on the NTU RGB+D 60 dataset, using cross-subject (CS) and cross-view (CV) protocols. Our model, based on EffB7, achieves competitive performance with 91.5% accuracy for CS and 94.8% accuracy for CV while relying solely on RGB data. This is significant, as many of the models that incorporate additional data sources (e.g., 3D-pose) achieve higher results. For instance, VPN++ + 3D-Poses (RNX3D101), which uses both RGB and 3D-pose data, achieves the highest accuracies of 96.6% (CS) and 99.1% (CV). However, the fact that our architecture can achieve comparable results using only the RGB modality demonstrates its efficiency and effectiveness, especially considering that it does not require additional inputs, such as 3D-pose data, which may demand more resources and specialized sensors.

In the comparative evaluation, the utilization of both 3D-pose and RGB modalities achieved higher performance than using RGB alone. This highlights the potential advantages of integrating multiple modalities to enhance performance, though it does require acquiring additional modalities, which may not always be feasible in practical situations. The challenge remains in the effort to improve research in the RGB-only domain to reach the performance achieved with multimodality approaches. Furthermore, incorporating multi-view inputs yields higher accuracy compared to a single-view input as it provides additional perspectives in action recognition. Considering fusion methods, the key factor is the selection of the appropriate layer for merging each model, as this choice directly impacts the performance.

The ability of our unified architecture to perform at this level with just RGB frames highlights its practicality for real-world applications where collecting 3D-pose data might not be feasible. This positions our approach as a competitive alternative to models relying on multimodality inputs.

The results in Table 4 showcase the performance of various SOTA models on the NTU RGB+D 120 dataset using CS1 and CS2 protocols, while our proposed MLF method achieves competitive accuracy, it falls slightly short when compared to Glimpse Clouds and DVAnet due to several factors. In comparison with Glimpse Clouds, our model does not incorporate sequential models like GRUs (Gated Recurrent Units), which are effective at capturing temporal dependencies across frames but increase computational overhead. Additionally, we process only five frames per input sample, whereas Glimpse Clouds utilizes eight frames, allowing it to extract more comprehensive temporal information from the data. In comparison with DVAnet, our model does not utilize transformer-based mechanisms, which are highly effective for modeling spatio-temporal data. Furthermore, our approach employs single-view training, focusing on a practical and efficient application scenario. In contrast, DVAnet adopts a multi-view training strategy, enabling it to leverage diverse perspectives of the action sequences, but at the cost of an increased number of cameras in practice. Despite these differences, our method provides a balanced trade-off between real-world usability and performance, demonstrating its potential for real-world applications.

The confusion matrix presented in Figure 3 illustrates the performance of our proposed method for each action class. It highlights challenges with distinguishing between action classes that involve similar movements. In the cross-subject (CS) evaluation, as seen in the zoomed-in section, class 16 (put on a shoe) is misclassified as class 17 (take off a shoe) about 20% of the time, due to the similarity in arm movements. In the cross-view (CV) evaluation, class 34 (rub two hands) and class 10 (clapping), which also exhibit similar motions, are misidentified roughly 20% of the time, particularly in the far-left and far-right corners of the matrix.

Another challenge arises in two-person classes where one individual remains inactive. In such cases, if the inactive person is not clearly visible in the OF images, the resulting OF may resemble that of a one-person class. This similarity can lead to misclassification. These results show the difficulty of accurately recognizing actions, especially when movements are similar or when the OF image of a minimally active person is not clearly distinguishable.

## 5. Conclusions

This paper introduces a multi-level fusion (MLF) architecture using EfficientNet-B7 as a case study, which leverages early, intermediate, and late fusion strategies to jointly capture spatial and temporal information. The proposed method exclusively uses RGB data as a single input modality, which is more practical for real-world applications, while models utilizing 3D-pose data generally achieve higher performance than those relying solely on RGB data, obtaining 3D skeleton data and additional modalities may be impractical or unfeasible in practical scenarios.

Our proposed architecture is simple yet effective in fusing information through early, intermediate, and late fusion. Early fusion combines contextual information from RGB images with temporal data from OF. Using EfficientNet-B7 as a case study, the model demonstrated competitive performance in the single-view RGB modality on the NTU RGB+D dataset, achieving the highest accuracy in the CS setup and only slightly lower accuracy compared to the best performance in the CV setup. We note that the proposed MLF architecture is flexible and can integrate different models, as illustrated in Figure 1.

Our proposed model utilizes EfficientNet-B7 as the backbone, comprising approximately 64 million parameters. Each modality is trained separately, and the softmax scores are combined at the final stage. If all streams were integrated during training, the model’s size would increase to approximately 150 million parameters. However, our approach faces certain limitations, particularly with more challenging datasets like NTU-RGB+D 120, where the number of classes is significantly larger and more complex. Additionally, the pre-trained pose estimation model used in our method was not specifically trained on the dataset, potentially limiting its ability to accurately detect human joints. Furthermore, the absence of an attention mechanism in our architecture makes it more challenging to classify larger and more intricate class sets effectively.

In future work, a comprehensive study of the guidelines for merging intermediate fusion across various models could be investigated. Furthermore, exploring the possibility of replacing the existing streams with different modalities, or incorporating additional streams, could enable the architecture to capture a wider range of information and improve its performance. Intermediate fusion requires the selection of the appropriate layer for merging each model, while the results of our experiments seem limited, they show potential for further investigation and improvement.

## Figures and Tables

**Figure 1 jimaging-10-00320-f001:**
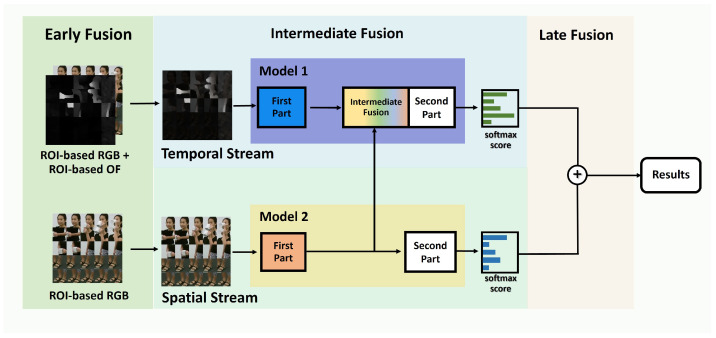
The proposed architecture integrates multi-level fusion through early, intermediate, and late fusion techniques. Early fusion is applied in the temporal stream, enriching the ROI-based OF with additional information from the ROI-based RGB frames. In addition, the spatial stream uses only ROI-based RGB frames as input. Intermediate fusion is used to merge extracted features from the spatial stream into the temporal stream, while late fusion is used to combine the softmax scores from both streams.

**Figure 2 jimaging-10-00320-f002:**
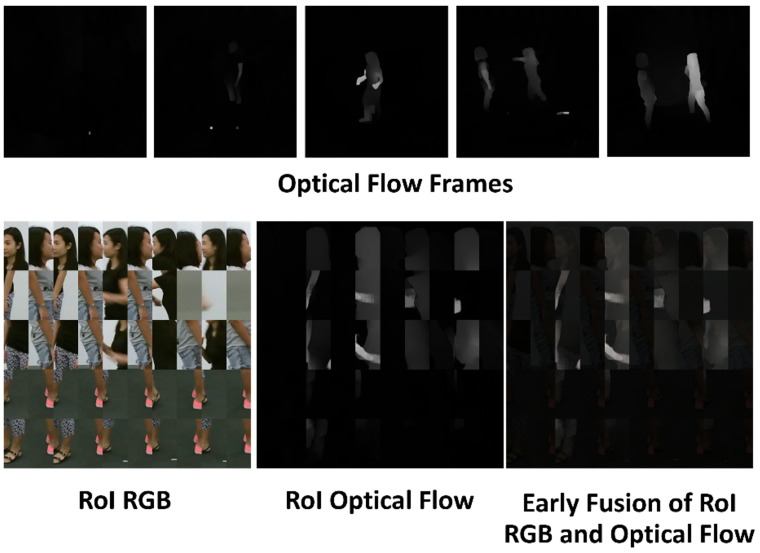
The top row presents five examples of OF frames extracted from pairs of selected video frames. The second row illustrates the corresponding ROI RGB frames, ROI OF frames, and the result of early fusion combining the ROI RGB and ROI OF for the same five frames. This demonstrates a soft shading effect, highlighting the motion of people while also showing non-moving parts in the images.

**Figure 3 jimaging-10-00320-f003:**
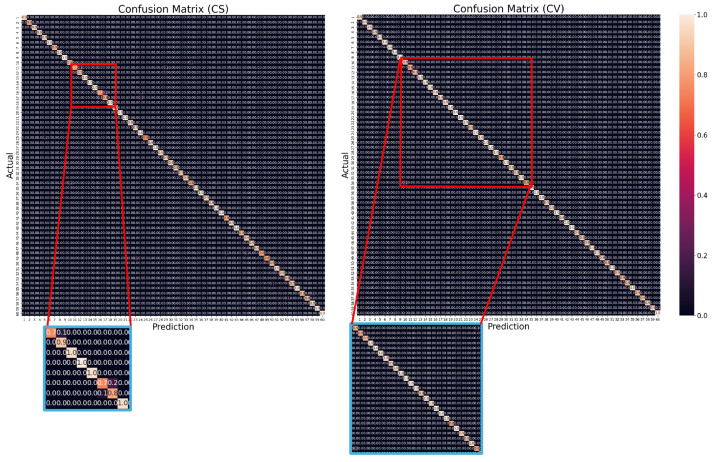
The confusion matrix for the proposed architecture on the NTU RGB+D 60 dataset, displaying actual classes on the vertical axis and predicted classes on the horizontal axis for both the cross-subject (CS) and the cross-view (CV) protocols, respectively.

**Table 2 jimaging-10-00320-t002:** Accuracy comparison of spatio-temporal intermediate fusion methods in EfficientNet-B7 using the CS protocol. The table outlines various fusion techniques and block-level combinations from both streams.

Intermediate Fusion	Accuracy (%)			
	**EffB7(S) (4)** **EffB7(T) (4)**	**EffB7(S) (5)** **EffB7(T) (5)**	**EffB7(S) (6)** **EffB7(T) (6)**	**EffB7(S) (7)** **EffB7(T) (7)**
Sum	82.03	85.20	87.69	85.89
Max	86.59	86.56	86.93	84.38
Concatenation	83.73	85.31	85.20	84.33
Convolution	83.93	86.14	88.46	85.95

**Table 3 jimaging-10-00320-t003:** Accuracy comparison with state-of-the-art approaches on the NTU RGB+D 60 dataset using CS and CV protocols. Our MLF method combines EffB7 (ST) and EffB7 (T), where “EffB7 (ST)” refers to the late fusion of spatio-temporal streams, with intermediate features fused at the 6th block of EfficientNet-B7 using convolution, and “EffB7 (T)” denotes the temporal EffB7 stream. Bold italics represent the highest accuracy for Pose + RGB, and bold values highlight the highest accuracy for RGB under each protocol.

Model	Pre-Trained Weights on RGB Backbone	Data Source		Accuracy (%)	
		**3D-Pose**	**RGB**	**CS**	**CV**
MMNet (EffB7) [9]	ImageNet	**x**	**x**	96.0	98.8
VPN (RNX3D101) [33]	ImageNet	**x**	**x**	95.5	98.0
VPN++ + 3D Poses (RNX3D101) [33]	ImageNet	**x**	**x**	* **96.6** *	* **99.1** *
LA-GCN [34]	**-**	**x**		93.5	97.2
SI-MM (RGB + 3D-Pose) [23]	**-**	**x**	**x**	92.6	97.9
P-CNN [35]	ImageNet		**x**	53.8	61.7
C3D [23,36]	ImageNet		**x**	63.5	70.3
Glimpse Clouds [37]	ImageNet		**x**	86.6	93.2
TSN [23,38]	ImageNet		**x**	88.5	90.4
SI-MM (RGB) [23]	**-**		**x**	90.7	96.3
DVAnet [39]	**-**		**x**	**93.4**	**98.2**
Our MLF (EffB7 (ST), EffB7 (S))	ImageNet		**x**	91.5	94.8

**Table 4 jimaging-10-00320-t004:** Accuracy comparison with state of the art approaches on the NTU RGB+D 120 dataset using cross-subject (CS1) and cross-setup (CS2) protocols. Our MLF method combines EffB7 (ST) and EffB7 (T), where “EffB7 (ST)” refers to the late fusion of spatio-temporal streams, with intermediate features fused at the 6th block of EfficientNet-B7 using convolution, and “EffB7 (T)” denotes the temporal EffB7 stream. Bold values indicate the highest accuracy for RGB under each protocol.

Model	Pre-Trained Weights on RGB Backbone	Accuracy	
		**CS1**	**CS2**
Two-Stream [10,40]	-	58.5	54.8
MMnet No joint weights [9]	ImageNet	67.2	71.1
I3D [40,41]	ImageNet	77.0	80.1
Glimpse Clouds [37,42]	ImageNet	83.8	83.5
DVAnet [39]	-	**90.4**	**91.6**
Our MLF (EffB7 (ST), EffB7 (S))	ImageNet	81.1	83.3

## Data Availability

Data Availability Statement: The NTU RGB+D dataset is openly available at https://rose1.ntu.edu.sg (accessed on 1 June 2022), references number [25,26].

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
