# Peer review of "J. Imaging2024, 10(12), 320;https://doi.org/10.3390/jimaging10120320"

_2313-433X, 2024, doi:10.3390/jimaging10120320_

Round 1

Reviewer 1 Report

Comments and Suggestions for Authors

Comments and suggestions

1.   This paper does a good job of highlighting the key elements of the research - using a multi-level fusion approach to combine multiple modalities (2D skeletons, optical flow) from a single RGB data source, and showing improved performance over single-modality models.

2.   It's clear the main focus is on enhancing human action recognition using multimodal techniques, which is an important and relevant problem.

3.   The mention of the challenges of collecting multiple data types in real-world applications is a nice acknowledgment of a practical consideration.

4.   The results showing 91.5% accuracy and comparability to state-of-the-art approaches are promising.

5.    Consider adding a bit more detail on the specific multi-level fusion approach used, if space allows.

6.    Quantify the performance improvement over single-modality baselines for more context.

7.   Discuss potential limitations or future work to provide a more well-rounded picture.

8.   For more completion, I would like to suggest the authors to include the following papers in their revised version.

(i) Swe Nwe Nwe Htun, Thi Thi Zin, Pyke Tin, “Image processing technique and hidden Markov model for an elderly care monitoring system”, Journal of Imaging, 2020, 6(6), 6060049.

Comments on the Quality of English Language

The paper is well-written. I found that the quality of English is good. 

Author Response

1. This paper does a good job of highlighting the key elements of the research - using a multi-level fusion approach to combine multiple modalities (2D skeletons, optical flow) from a single RGB data source, and showing improved performance over single-modality models.

Response: Thank you for your positive feedback. We appreciate that you found our work on the multi-level fusion approach and its improvements over single-modality models to be a valuable contribution.

2. It's clear the main focus is on enhancing human action recognition using multimodal techniques, which is an important and relevant problem.

Response: We are pleased that the focus on enhancing human action recognition using multimodal techniques was recognized as an important and relevant problem. This reinforces the motivation behind our research.

3. The mention of the challenges of collecting multiple data types in real-world applications is a nice acknowledgment of a practical consideration.

Response: Thank you for noting our acknowledgment of the challenges involved in collecting multiple data types in real-world applications. We believe this aspect is crucial in advancing practical implementations of multimodal systems.

4. The results showing 91.5% accuracy and comparability to state-of-the-art approaches are promising.

Response: We are glad you found the results promising.

5. Consider adding a bit more detail on the specific multi-level fusion approach used, if space allows.

Response: Thank you for your suggestion. We have expanded the description of the specific multi-level fusion approach used in our method in Line 287, detailing the architecture and integration techniques, in the revised manuscript. 

6. Quantify the performance improvement over single-modality baselines for more context.

Response: Thank you for this insightful comment. While we acknowledge the value of quantifying the performance improvement over single-modality baselines for additional context, we have included results from the NTU RGB+D 120 dataset to provide a broader evaluation. Due to the limited timeframe for revisions, we were unable to include the performance improvement analysis specifically for the NTU RGB+D 60 single-modality baselines since doing so would imply that we need to do it on NTU RGB+D 120 accordingly.

7. Discuss potential limitations or future work to provide a more well-rounded picture.

Response: Thank you for highlighting this. In the revised manuscript, we have included a discussion on the limitations of our approach in Line 497, such as the reliance on large computational resources for training and potential challenges in real-time deployment. We have also outlined future research directions in Line 507, including optimizing the model for lower resource environments and exploring more advanced fusion techniques.

8. For more completion, I would like to suggest the authors to include the following papers in their revised version.

(i) Swe Nwe Nwe Htun, Thi Thi Zin, Pyke Tin, “Image processing technique and hidden Markov model for an elderly care monitoring system”, Journal of Imaging, 2020, 6(6), 6060049.

Response: Thank you for the recommendation. We have reviewed the suggested paper,  and included it in Line 19 on its relevance in the context of human action recognition in healthcare. Its insights on Markov modeling provide valuable context for the interpretability of sequential data in our application.

Reviewer 2 Report

Comments and Suggestions for Authors

A practical study on multilevel feature fusion in the task of human action recognition is presented. Several known methods (YOLO, optical flows, some types of neural networks) are combined. The resulting architecture involves two channels and feature fusion. Although the study does not introduce new concepts or methods, it is an interesting example of solving a practical problem. There are the following drawbacks.

1. The authors claim their method has similar or better accuracy than the rivals. However this is not sufficient information to judge on its usability and merit. It is well known that bigger classifier (with more parameters) may be more accurate (sometimes due to dimension curse) than similar smaller one. Thus the authors should present number of parameters in their solution and its rivals, as well as training time and time of making decision (required amount of calculations).

2. Also the issues of training and testing data sets are not clear. Were the data separated in these sets, and how? For instance, were the persons separated so as the persons in testing set were never met in the training one?

Minor drawbacks.

1. Line 247. Types of Model 2 are not given, as it was for Model 1.

2. Formula (1). There is no need to define multiplication with any sign. Moreover, asterisk is used further (line 273) to denote convolution operation.

3. Line 273. All fusions before this are given in terms of tensor x. This one should also be given in this terms.

The paper can be published after correcting the outlined issues and repeated review.

Author Response

1. The authors claim their method has similar or better accuracy than the rivals. However this is not sufficient information to judge on its usability and merit. It is well known that bigger classifier (with more parameters) may be more accurate (sometimes due to dimension curse) than similar smaller one. Thus the authors should present number of parameters in their solution and its rivals, as well as training time and time of making decision (required amount of calculations).

Response

We acknowledge that larger models often tend to produce better results. However, regarding the parameters of the models used in our comparative analysis, most of the original papers did not disclose their parameter counts. Therefore, we estimated these parameters based on the backbones used. While we strove for accuracy, these estimates may not be entirely precise. As such, we decided not to include them in our paper. Nonetheless, we present our parameter estimates in this document for your reference.

However, our model parameters are provided in the paper in Line 497.

2. Also the issues of training and testing data sets are not clear. Were the data separated in these sets, and how? For instance, were the persons separated so as the persons in testing set were never met in the training one?

Response: Thank you for your suggestion. We provide additional details in Line 316.

3. Line 247. Types of Model 2 are not given, as it was for Model 1.

Response: Thank you for noting this omission. We have updated the manuscript to include the specific details and types of Model 2, ensuring consistency with the description provided for Model 1. We mentioned the types of both Model 1 and Model 2 in Line 199 before. However, we also included additional details (similar to Model 1) in Line 247.

4. Formula (1). There is no need to define multiplication with any sign. Moreover, asterisk is used further (line 273) to denote convolution operation.

Response: We agree with your observation. We have revised Formula (1) at the top of Line 256 by removing the unnecessary definition of multiplication. Additionally, we have changed the convolution operation in convolution fusion definition in the Line 276.

5. Line 273. All fusions before this are given in terms of tensor x. This one should also be given in this terms.

Response: We agree with your observation. We changed the notation according to your suggestion in Line 276.

Reviewer 3 Report

Comments and Suggestions for Authors

This manuscript under review presents a multi-level fusion (MLF) architecture based on EfficientNet-B7 to recognize human activities. The paper is well-structured and clear. The reviewer has the following comments.

- The novelty of the paper appears to be limited to experimenting with the early to late fusion strategies evaluation using EfficientNet-B7. The technical novelty is therefore limited. 

- Figure 1 can be improved by adding the details of the early and late fusion sections. The current diagram does not say much about these two parts. One detailed figure for each, early, intermediate, and late fusion scheme will be appreciated.

- The experimental evaluation is performed on NTU RGB+D 60 and it is commendable that the results are compared with other similar approaches. The results presented in Table 3 show a marginal gain in accuracy with the proposed scheme over the other methods. It, however, requires more experiments to confirm the gains reported in the table. It is suggested to test the proposed method on at least two more datasets, e.g., NTU-120, Toyota-Smarthome, UCI-HAR, and WISDM.

- Consider adding more recent techniques in comparative analysis, such as, Cheng, D., Zhang, L., Bu, C., Wu, H. and Song, A., 2023. Learning hierarchical time series data augmentation invariances via contrastive supervision for human activity recognition. Knowledge-Based Systems, 276, p.110789. Yang, Z., Li, Y. and Zhou, G., 2024. Unsupervised Sensor-Based Continuous Authentication With Low-Rank Transformer Using Learning-to-Rank Algorithms. IEEE Transactions on Mobile Computing. Nguyen, D.A., Pham, C. and Le-Khac, N.A., 2024. Virtual fusion with contrastive learning for single sensor-based activity recognition. IEEE Sensors Journal. Khan, M. H., Shafiq, H., Farid, M. S., & Grzegorzek, M. (2024). Encoding human activities using multimodal wearable sensory data. Expert Systems with Applications, 125564. Batool, S., Khan, M.H. and Farid, M.S., 2024. An ensemble deep learning model for human activity analysis using wearable sensory data. Applied Soft Computing, p.111599.

Author Response

1. The novelty of the paper appears to be limited to experimenting with the early to late fusion strategies evaluation using EfficientNet-B7. The technical novelty is therefore limited.

Response: We appreciate the reviewer’s observation. To put it in practical terms, our work aims to enhance how different types of information (like RGB data and pose data) are combined to recognize human actions more effectively. While the main focus of our research is on exploring and improving these fusion strategies, we also propose a unique way to merge different levels of information within the model. This proposed approach is specifically designed to improve the accuracy in real-world applications like video surveillance, gesture recognition, and activity tracking.

2. Figure 1 can be improved by adding the details of the early and late fusion sections. The current diagram does not say much about these two parts. One detailed figure for each, early, intermediate, and late fusion scheme will be appreciated.

Response: We appreciate the suggestion to add detailed figures for the early, intermediate, and late fusion schemes. However, we have decided not to include these additional figures due to time constraints and the need to focus on finalizing other critical aspects of the paper. While we agree that such diagrams could provide valuable insights, we believe the current description sufficiently conveys the core concepts of our approach. Future versions of this work may include these details to enhance clarity further. 

3. The experimental evaluation is performed on NTU RGB+D 60 and it is commendable that the results are compared with other similar approaches. The results presented in Table 3 show a marginal gain in accuracy with the proposed scheme over the other methods. It, however, requires more experiments to confirm the gains reported in the table. It is suggested to test the proposed method on at least two more datasets, e.g., NTU-120, Toyota-Smarthome, UCI-HAR, and WISDM.

Response: Thank you for your suggestion. We have additionally tested our model on the NTU-RGB+D 120 dataset, which features a significantly larger dataset and a greater number of classes. However, the results on NTU-RGB+D 120 did not surpass those of other state-of-the-art methods. The potential reasons for this are discussed in detail in Line 451.

4. Consider adding more recent techniques in comparative analysis, such as,

- Cheng, D., Zhang, L., Bu, C., Wu, H. and Song, A., 2023. Learning hierarchical time series data augmentation invariances via contrastive supervision for human activity recognition. Knowledge-Based Systems, 276, p.110789.  

- Yang, Z., Li, Y. and Zhou, G., 2024. Unsupervised Sensor-Based Continuous Authentication With Low-Rank Transformer Using Learning-to-Rank Algorithms. IEEE Transactions on Mobile Computing. 

- Nguyen, D.A., Pham, C. and Le-Khac, N.A., 2024. Virtual fusion with contrastive learning for single sensor-based activity recognition. IEEE Sensors Journal. 

- Khan, M. H., Shafiq, H., Farid, M. S., & Grzegorzek, M. (2024). Encoding human activities using multimodal wearable sensory data. Expert Systems with Applications, 125564. 

- Batool, S., Khan, M.H. and Farid, M.S., 2024. An ensemble deep learning model for human activity analysis using wearable sensory data. Applied Soft Computing, p.111599.

Response: We have included DVAnet (2023) in our comparative analysis for both NTU-RGB+D and NTU-RGB+D 120. The results indicate that DVAnet outperformed our models. However, this can be attributed to its use of an attention-based approach combined with multi-view training, which significantly enhances performance. Additional details on this comparison are provided in Line 451.

Reviewer 4 Report

Comments and Suggestions for Authors

The study proposed a multi-level fusion approach using only RGB data for human action classification, employing two models to extract temporal and spatial features and fusing the features at early, intermediate, and late stages. The proposed method based on EfficientNet-B7 achieved superior performance according to the cross-subject protocol than any other compared state-of-the-art methods also using only RGB data, assuming the literature review was comprehensive. The study results are worth reporting. The manuscript is concisely written and explains the methodology sufficiently. The study is properly justified. I did not observe any major issues with the manuscript.

Author Response

1. The study proposed a multi-level fusion approach using only RGB data for human action classification, employing two models to extract temporal and spatial features and fusing the features at early, intermediate, and late stages. The proposed method based on EfficientNet-B7 achieved superior performance according to the cross-subject protocol than any other compared state-of-the-art methods also using only RGB data, assuming the literature review was comprehensive. The study results are worth reporting. The manuscript is concisely written and explains the methodology sufficiently. The study is properly justified. I did not observe any major issues with the manuscript.

Response: Thank you for your feedback. We appreciate your recognition of our proposed multi-level fusion approach using EfficientNet-B7. We are glad that you found the manuscript concisely written and the methodology sufficiently explained.

Round 2

Reviewer 2 Report

Comments and Suggestions for Authors

The authors have corrected the issues outlined in the comments. The paper can be published now.

Reviewer 3 Report

Comments and Suggestions for Authors

The authors adequately addressed my concerns about the previous version of the manuscript, so the paper can be considered for acceptance. However, the related work section must be enriched with the latest papers published on this topic, e.g.,  Tazeem Haider, et al., "An Optimal Feature Selection Method for Human Activity Recognition Using Multimodal Sensory Data". Information, 15(10), 593, 2024, Dentamaro, Vincenzo, Vincenzo Gattulli, Donato Impedovo, and Fabio Manca. "Human activity recognition with smartphone-integrated sensors: A survey." Expert Systems with Applications (2024): 123143.,  Guo, Changru, Yingwei Zhang, Yiqiang Chen, Chenyang Xu, and Zhong Wang. "Modality Consistency-Guided Contrastive Learning for Wearable-Based Human Activity Recognition." IEEE Internet of Things Journal (2024)., Muhammad Hassan Khan, Hadia Shafiq, Muhammad Shahid Farid, Marcin Grzegorzek, "Encoding human activities using multimodal wearable sensory data", Expert Systems with Applications, Volume 261, 2025, 125564, Nazish Ashfaq, et al, "Optimizing Human Activity Recognition with Ensemble Deep Learning on Wearable Sensor Data," International Journal of Innovations in Science & Technology, vol. 6, No. 4, pp. 1751-1767, 2024.